# MF-Match: A Semi-Supervised Model for Human Action Recognition

**DOI:** 10.3390/s24154940

**Published:** 2024-07-30

**Authors:** Tianhe Yun, Zhangang Wang

**Affiliations:** School of Information and Communication Engineering, Beijing Information Science and Technology University, Beijing 100101, China; 2022020504@bistu.edu.cn

**Keywords:** cross domain, human action recognition, radar, semi-supervised learning

## Abstract

Human action recognition (HAR) technology based on radar signals has garnered significant attention from both industry and academia due to its exceptional privacy-preserving capabilities, noncontact sensing characteristics, and insensitivity to lighting conditions. However, the scarcity of accurately labeled human radar data poses a significant challenge in meeting the demand for large-scale training datasets required by deep model-based HAR technology, thus substantially impeding technological advancements in this field. To address this issue, a semi-supervised learning algorithm, MF-Match, is proposed in this paper. This algorithm computes pseudo-labels for larger-scale unsupervised radar data, enabling the model to extract embedded human behavioral information and enhance the accuracy of HAR algorithms. Furthermore, the method incorporates contrastive learning principles to improve the quality of model-generated pseudo-labels and mitigate the impact of mislabeled pseudo-labels on recognition performance. Experimental results demonstrate that this method achieves action recognition accuracies of 86.69% and 91.48% on two widely used radar spectrum datasets, respectively, utilizing only 10% labeled data, thereby validating the effectiveness of the proposed approach.

## 1. Introduction

HAR technology has garnered significant interest from academia and industry due to its broad applicability across various social domains including marketing, sports, fitness, and eldercare. HAR utilizes diverse input data modalities such as visible images, 3D skeleton data, depth images, WiFi signals, and radar signals [1]. Radar signals stand out among these modalities for their superior ability to safeguard personal privacy and their resilience to environmental changes like lighting conditions. Moreover, radar technology can penetrate obstacles such as walls, enabling HAR to capture human activities that might otherwise be obscured, thereby expanding its range of applications.

To date, researchers have conducted extensive investigations into HAR using radar signals, yielding notable outcomes. Nevertheless, the scarcity of precisely labeled radar feature map data, exacerbated by the high costs associated with data labeling, often fails to satisfy the expansive requirements of deep learning models for extensive training datasets. Consequently, the potential for enhancing model performance remains constrained. Thus, addressing the challenge of effectively leveraging limited labeled radar data to train deep learning networks for achieving high-precision HAR stands as a pivotal issue in this domain.

To tackle this issue, researchers have introduced a training framework centered on transfer learning (TL). Transfer learning involves leveraging large-scale datasets (referred to as source-domain data) to pretrain models for acquiring general data representations. Subsequently, these pretrained models, equipped with prior knowledge, are fine-tuned through supervised learning on smaller datasets from a target domain. This approach aims to enhance model performance on specific tasks within the target domain. Park et al. [2] conducted a study using AlexNet [3] and VGG16 [4], initially pretrained on the large-scale natural image dataset ImageNet [5] and then fine-tuned on radar feature map datasets. They achieved an 80.3% accuracy in HAR. However, the substantial disparity in feature distributions between radar feature maps and commonly used large-scale datasets of natural images poses a significant challenge to further enhancing transfer learning effectiveness. In addition to pretraining strategies, alternative approaches such as unsupervised domain adaptation algorithms have been proposed. Du et al. [6] employed adversarial learning to generalize feature extractors from a source to a target domain, aiming to maximize domain discriminator classification errors and enhance inter-domain feature invariance. They suggested replacing ImageNet with the MOCAP behavioral capture dataset as source-domain data to minimize feature distribution gaps with target-domain data (i.e., radar feature maps of human behavior). However, these unsupervised methods face difficulties in achieving robust action recognition accuracy due to the absence of labeled data to guide the learning process.

To maximize the utilization of both limited labeled data and larger sets of unlabeled data, this study adopts a model training framework centered on semi-supervised learning (SSL) [7]. Unlike purely supervised methods that solely rely on labeled data, semi-supervised learning offers a strategy to uncover underlying data distribution patterns from unlabeled samples [8]. This approach mitigates the necessity for extensive labeling efforts and effectively reduces data collection costs. Moreover, compared to unsupervised learning approaches, semi-supervised learning leverages available labeling information to steer the learning process, facilitating the acquisition of more discriminative feature representations by the model.

Drawing inspiration from the success of SSL, this study introduces MF-Match, an innovative semi-supervised deep learning algorithm for radar signal-based HAR. This approach leverages cost-effective unlabeled samples to complement the limited labeled radar feature maps with the aim of enhancing the accuracy of human behavior classification. The algorithm employs comparative learning to initially classify unlabeled data, thereby refining the accuracy of the generated pseudo-labels. To mitigate the challenge posed by radar feature maps exhibiting high similarity across different human behaviors, diverse transformations are applied to unlabeled samples. These transformations amplify subtle feature differences while preserving the original semantics, thereby improving action recognition accuracy. Additionally, weight sharing of the encoder network between labeled and unlabeled data is implemented, enhancing pseudo-label accuracy while reducing model parameters and improving overall computational efficiency.

The main contributions of this paper are as follows:(1)This paper introduces a radar signal-based human action recognition algorithm that utilizes a semi-supervised learning framework. This approach aims to diminish the algorithm’s reliance on extensive labeled Raytheon data by extracting discriminative features from unlabeled data.(2)Addressing the challenge of distinguishing radar signals across various behavior classes, this study proposes a comparative learning-based pseudo-label generation method. This method enhances the accuracy of human behavior recognition by implementing multiple strategies to magnify feature distinctions between classes.(3)In experiments conducted on a publicly accessible radar feature map dataset, the method proposed in this paper demonstrates a human action recognition accuracy of 91%. This outperforms existing methods utilizing supervised, unsupervised, and semi-supervised learning frameworks, especially noteworthy given that the training data include only 10% labeled samples. Furthermore, numerous ablation experiments corroborate the effectiveness of the strategies proposed herein, including the enhancement of inter-class feature distinctions and the sharing of encoder weights.

## 2. Related Work

The proposed framework is mainly related to three techniques: (1) cross-domain human action recognition, (2) semi-supervised learning, and (3) consistency regularization.

A.Cross-domain Human Action Recognition

In recent years, radar-based human sensing technology has garnered considerable interest due to significant advancements. Given the pivotal role of HAR in human–computer interaction, extensive research has been conducted to detect human movements using radar signals. To alleviate the challenges of data collection and labeling and advance recognition models, cross-domain human action recognition has emerged as a prominent research focus. This research is broadly categorized into two main methods: millimeter wave-based and continuous wave-based approaches.

In the domain of continuous wave-based research, Hernang’ omez and colleagues [9] introduced a dual-stream CNN architecture (multibranch CNN) designed to process micro-Doppler and distance spectrograms as inputs, aiming to characterize structural features of targets. They simultaneously computed the distance and Doppler spectrograms by aggregating corresponding axes, thereby alleviating the memory and computational burden associated with additional distance information. In contrast, Wang et al. [10] employed a stacked recurrent neural network (RNN) combined with a long short-term memory (LSTM) unit. This model utilized feature maps derived from raw radar data inputs to capture time-varying Doppler and micro-Doppler signals, which served as features of human body motion for subsequent human behavior classification. Additionally, W. Li et al. [11] proposed the Doppler and distance decision level convergence network model (DRCNet), which effectively learned DT and RT maps to achieve outstanding recognition performance. Furthermore, W. Li et al. [12] utilized the integration of FMCW signals and cameras to mitigate environmental influences on recognition accuracy through multisignal fusion processing.

Despite achieving respectable performance, all aforementioned models still require improvement, primarily due to high costs and limited advancements. This limitation stems from insufficient labeled data, preventing these methods from fully leveraging information within the target domain data to optimize recognition models. In response, this study introduces a semi-supervised approach for HAR. The proposed model aims to effectively harness unlabeled data from the target domain along with limited labeled features to enhance recognition model performance within the target domain.

B.Semi-supervised learning

Semi-supervised learning (SSL) leverages a limited set of labeled data alongside a substantial volume of unlabeled data to enhance model performance. Its primary distinction from supervised learning lies in the quantity of manually labeled data required for training. While supervised learning demands a significant amount of labeled data to teach the model the relationship between inputs and output labels, SSL integrates a small set of labeled data with a large pool of unlabeled data to train the network. Although SSL may sacrifice some timeliness or accuracy compared to supervised learning, it notably reduces the burden of costly labeling processes.

In recent developments, several effective SSL methods have emerged, such as MixMatch [13], FixMatch [14], and ReMixMatch [15], all rooted in the data augmentation paradigm. Among these, MixMatch [13] assigns low-entropy labels to augmented instances from unlabeled data and integrates a combination of labeled and unlabeled data within SSL. FixMatch [14], on the other hand, utilizes the model’s predictions on lightly augmented unlabeled images to generate pseudo-labels. ReMixMatch [15] generates pseudo-labels through weak augmentation and enforces strong consistency across instances, thereby enhancing model robustness and performance.

In the realm of HAR, an increasing number of researchers are turning their attention to the application of SSL to mitigate the costs associated with data labeling. For instance, Campbell and Ahmad [16] introduced a semi-supervised attention enhancement model (AA-CAE) for radar-based HAR. The model underwent initial pretraining followed by fine-tuning using 20% labeled data, ultimately achieving a classification accuracy of 75%. In contrast, Rahman and Gurbuz [17] devised a self-supervised comparative learning framework leveraging multiresolution micro-Doppler and physics-aware GAN for radar data augmentation. They employed the consistency principle to fine-tune the model using 20% labeled radar data, achieving an accuracy of 88%. Additionally, X. Li et al. [18] proposed a radar-based HAR semi-supervised transfer learning algorithm, joint domain semantic transfer learning (JDS-TL), which achieved an accuracy of 87.6% with only 10% labeled data.

C.Consistency regularization

Consistency regularization has become an integral component of SSL models, grounded in smoothing and clustering assumptions and leveraging unlabeled data to enhance model performance. Specifically, it mandates that data points with different labels should reside in low-density regions, while maintaining similar outputs for data points with labels akin to them even under perturbation. This concept was initially introduced in [19] through the semi-supervised learning method PEA, which stresses the necessity of preserving consistency across all intermediate representations amidst input perturbations. Building upon this, the Π-model algorithm detailed in [20] further refines consistency regularization principles. It integrates traditional data augmentation techniques like translation, rotation, or random dropout to augment the model’s recognition capabilities. In [21], researchers introduced the innovative concept of feature consistency, arguing that features within the same data category should exhibit coherence. To achieve this, the study adopted pseudo-labeling in unsupervised learning, establishing coherence among features of identical categories.

## 3. Har Based on Semi-Supervised MF-Match

To enhance the accuracy of human motion recognition based on radar signals under conditions of limited labeled data, a novel semi-supervised learning framework is proposed in this paper. This section introduces the proposed MF-Match method and provides a detailed description of its components.

### 3.1. Problem Setup

We assume that there is a radar feature map dataset $S=S_{L}+S_{U}$. $S_{L}={\left\{\left(x_{i},y_{i}\right)\right\}}_{i=1}^{L}$ represents the set of labeled data, which contains a total of L pairs of samples, where $x_{i}$ denotes the $i^{th}$ input feature map, and $y_{i}=\left[y_{i}^{1},...,y_{i}^{C}\right]\subseteq {\left\{0,1\right\}}^{C}$ is the one-hot category label (C represents the total number of action categories). Similarly, $S_{U}={\left\{\left(x_{u}\right)\right\}}_{u=1}^{U}$ represents the set of unlabeled radar feature maps (U≫L) in the dataset S, with the same category distribution as $S_{L}$. MF-Match is designed to mine the feature information embedded in the unlabeled data $S_{U}$ to assist in improving the overall human action classification performance. The model architecture and key techniques will be described in the following subsections.

### 3.2. HAR Pipeline with MF-Match

Here, we briefly outline the main steps in the HAR solution pipeline using MF-Match. In the next section, we provide a detailed explanation of the key techniques employed in this method.

**1. Radar data preprocessing:** To enable human action recognition based on radar signals using the depth model, we begin by processing the Doppler spectral extraction of raw radar signals, which capture human body echo signals as illustrated in Figure 1. Initially, we perform a 256-point FFT on the raw radar data and apply a rectangular window function to shift the resultant frequency domain data (Figure 1a). Following this step, we employ an infinite impulse response filter (IIR) to eliminate low-frequency noise from the data (Figure 1b). Subsequently, we utilize the joint time–frequency transform [22,23,24] to extract time-varying frequency information crucial for effective identification. Specifically, we apply the STFT with a 95% window overlap, a four-point fill factor, and a Doppler resolution of 1.25 Hz, which is normalized to convert the noise-reduced radar signals into a two-dimensional time-domain Doppler spectrogram (Figure 1c). In this study, this Doppler spectrogram is denoted as the “radar feature map” and serves as the input data for the proposed human action recognition classification model.

**2. Using MF-Match for human action recognition:** The overall framework of the model of MF-Match is shown in Figure 2, which contains the following four main modules: supervised learning module, comparison learning module, self-supervised learning module, and pseudo-label matching module.

For the labeled radar feature dataset $S_{L}$, MF-Match adopts the supervised learning method, which utilizes the cross-entropy loss to train the encoder $E_{Sup,B}$ containing the convolutional encoder and the fully connected projection header.

For the unlabeled radar feature map $S_{U}$, MF-Match employs both contrast learning and self-supervised learning methods to mine the human behavior-related features embedded in it. In the contrast learning module, the model first processes the data in $S_{U}$ with two kinds of weak augmentation (${Aug}_{w}$, ${Aug^{\prime }}_{w}$) and then utilizes the momentum encoder $E_{Con,M}$ [25] to perform the contrast learning and finally obtains the corresponding pseudo-labels. At the same time, we use a sharing mechanism to share the parameters of $E_{Sup,B}$ and $E_{Con,B}$ to utilize the parameters of labeled data to improve the utilization efficiency of unlabeled data. In addition to this, MF-Match also performs strong augmentation ${Aug}_{s}$ on the unlabeled data in $S_{U}$ to enhance the diversity of the data, and based on this, the self-supervised algorithm is utilized to extract a more robust feature representation. The final classification prediction is performed by the momentum encoder $E_{Self,M}$, and these results are matched with the pseudo-labels obtained in the comparative learning module through the cross-entropy loss $L_{U}$, which is iterated to optimize the model parameters and improve the classification accuracy.

## 4. Key Technologies of MF-Match

### 4.1. Inputs and Outputs

We first selected B labeled data ($x_{i}$,$y_{i}$) from the labeled human action radargram dataset $S_{L}$ and first selected γB unlabeled data $x_{u}$ from the unlabeled dataset $S_{U}$ (γ is a hyperparameter). Subsequently, we augmented the data with two different levels of augmentation, including strong and weak augmentation, to obtain all augmented data $x^{A}ug$, where the augmentation of labeled data is ${x_{i}}^{Aug_{w}}$, and the weak and strong augmentation of unlabeled data are ${x_{u}}^{Aug_{w}}$ and ${x_{u}}^{Aug_{s}}$, respectively. We defined the predicted value of x after passing through the model as $\hat{y}$. Therefore, the predicted value of $x_{i}$ after weak enhancement is ${\hat{y_{i}}}^{Aug_{w}}$, and the predicted value of $x_{u}$ after strong and weak enhancement is ${\hat{y}}_{u}^{{Aug}_{s}}$ and ${\hat{y}}_{u}^{{Aug}_{w}}$, respectively.

MF-Match employs data augmentation to enrich the dataset and enhance classification accuracy. During training, labeled feature maps undergo mild augmentation, while unlabeled feature maps undergo both mild and intensive augmentation. Mild augmentation involves simple transformations like color adjustments and shifts, which are gentle and ensure stability and reliability of generated pseudo-labels. This approach allows the model to learn while preserving original sample features and improving pseudo-label quality. In contrast, intensive augmentation utilizes the advanced technique of RandAugment [26]. RandAugment enhances data by randomly selecting multiple image transformations, ensuring diversity without compromising important feature information. Specifically, only two transformations are randomly chosen to prevent excessive alterations, thereby enhancing data diversity and maintaining stable model performance across varied input transformations.

### 4.2. Supervised Learning Module

For the labeled data, since ${\hat{y}}_{i}^{Aug_{w}}$ corresponds to the ground truth $y_{i}$, we can utilize the cross-entropy loss function formula to compute the supervised learning loss $L_{x}$: (1)$$L_{x}=\frac{1}{B}\sum_{i=1}^{B}H(y_{i},{\hat{y}}_{i}^{Aug_{w}})$$
where $H\left(•\right)$ denotes the cross-entropy function. The labeled dataset $S_{L}$ is employed to train the classification model using the supervised loss function $L_{x}$. $L_{x}$ constitutes a component of the objective function *L*, which is used to compute gradients and update parameters during the training process. Additionally, unsupervised learning techniques, including contrastive learning, pseudo-labeling, and consistency regularization, are incorporated to effectively leverage additional data.

### 4.3. Unsupervised Learning Module

#### 4.3.1. Generating Pseudo-Labels

In order to effectively utilize the data from unlabeled radargrams, we pseudo-label the unlabeled data by generating pseudo-labels. Specifically, sample labeling is performed by means of the value τ [27]. The pseudo-labeling can be expressed as
(2)$$y_{u}^{j}=\left\{\begin{array}{c} \;1,\;{\hat{y}}_{u}^{Aug_{w}\left(j\right)}\geq \tau \\ \;0,\;otherwise \end{array}\right.j=1,2,\ldots ,C$$
where *C* denotes the number of human action categories. The cross-entropy loss function is used to represent the self-supervised loss function $L_{self}$ as
(3)$$L_{self}=\frac{1}{\gamma B}\sum_{j=1}^{\gamma B}H(y_{u}^{j},{\hat{y}}_{u}^{Aug_{w}\left(j\right)})$$
where γB is the number of pseudo-labels, and ${\hat{y}}_{u}^{{Aug}_{w}\left(j\right)}$ denotes its prediction after enhancement. This model does not directly use $L_{self}$ for model optimization here but combines it with consistency regularization in order to reduce the computational cost.

#### 4.3.2. Consistency Regularization

Using only pseudo-labeling in semi-supervised learning can initially introduce confirmation bias with model-generated pseudo-labels, leading to a misguided training process that undermines model performance. In contrast, consistency regularization enhances the model’s predictive reliability on unlabeled data by integrating a loss term during training. This approach involves applying diverse perturbations to input data to enforce consistent model outputs under varying conditions. Initially addressed by adding noise, Xie [28] et al. demonstrated that employing data augmentation enhances the effectiveness of consistency regularization, leading to improved model performance. Therefore, in this framework, we utilize both mild and intensive augmentation to diversify image styles and enhance model performance. These augmented datasets are input into the classification model, which is optimized using cross-entropy, specifically the consistency regularization loss function $L_{co-re}$ between predictions. The consistency regularization loss function $L_{co-re}$ is defined as
(4)$$L_{co-re}=\frac{1}{\gamma B}\sum_{j=1}^{\gamma B}H({\hat{y}}_{u}^{Aug_{w}\left(j\right)},{\hat{y}}_{u}^{Aug_{s}\left(j\right)})$$

#### 4.3.3. Matching Loss Function

While both the self-supervised loss function $L_{self}$ and the consistency regularization loss function $L_{co-re}$ effectively utilize unlabeled data, their direct application can significantly increase computational resource consumption. This can limit the efficiency of hyperparameter optimization and parallel computation, thereby indirectly impacting the accuracy of the classification model. To address this challenge, we integrate these two loss functions based on their common components, specifically ${\hat{y}}_{u}^{{Aug}_{w}\left(j\right)}$, merging them into a unified loss function termed the matched cross-entropy loss $L_{u}$, illustrated in Figure 3: (5)$$L_{m}=\frac{1}{\gamma B}\sum_{j=1}^{\gamma B}H(y_{u}^{j},{\hat{y}}_{u}^{Aug_{s}\left(j\right)})$$

#### 4.3.4. Comparing Loss Functions

Pseudo-labeling suffers from the problem of confirmation error, which is mainly caused by the model giving wrong pseudo-labeling. To be able to mitigate this problem, i.e., to improve the accuracy of pseudo labeling, we use the contrast loss function $L_{c}$. In the proposed framework, we use two encoders: a basic encoder $\theta_{q}$ and a momentum encoder $\theta_{k}$ encoding. The structure of the momentum encoder is the same as that of the basic encoder, and its parameter update strategy satisfies the following equation: (6)$$\theta_{k}\left(•\right)=m\theta_{q}+\left(1-m\right)\theta_{k}$$
where m∈(0,1) is the momentum coefficient. The structure is composed of a convolutional encoder θ(•) and a projection head. The output vectors are $q_{1}$, $q_{2}$, $k_{1}$, and $k_{2}$, where $q_{1}$ and $q_{2}$ stand for “query” and $k_{1}$ and $k_{2}$ stand for “key”. Two minimization contrast loss functions [29] are used, each of which is represented in the form of InfoNCE [30]: (7)$$L_{q}=-\log\frac{e^{\left(q\cdot k^{+}/\tau \right)}}{e^{\left(q\cdot k^{+}/\tau \right)}+\sum_{k^{-}}e^{\left(q\cdot k^{-}/\tau \right)}}$$
where $k^{+}$ is the output of $f_{k}$ on the same image as *q*, i.e., the positive samples of *q*. The set $k^{-}$ is the other outputs of $f_{k}$, i.e., the negative samples of *q*. τ is an L2-normalized with temperature hyperparameter, and the final contrast loss function is
(8)$$L_{c}=-\log\frac{e^{\left(q_{1}\cdot {k_{2}}^{+}+q_{2}\cdot k_{1}^{+}/\tau \right)}}{\left(e^{\left(q_{1}\cdot {k_{2}}^{+}/\tau \right)}+\sum_{{k_{1}}^{-}}e^{\left(q_{1}\cdot {k_{2}}^{-}/\tau \right)}\right)\cdot \left(e^{\left(q_{2}\cdot {k_{1}}^{+}/\tau \right)}+\sum_{{k_{2}}^{-}}e^{\left(q_{2}\cdot {k_{1}}^{-}/\tau \right)}\right)}$$

### 4.4. Objectives and Training

The final objective function L is composed of a supervised loss function $L_{x}$, a matching loss function $L_{m}$, and a comparison loss function $L_{c}$. The objective function L is
(9)$$L=L_{x}+\eta L_{m}+\mu L_{c}$$
where η and μ are the weights of the loss function. During the training process, we calculate and minimize the objective function to update the parameters of the neural network according to SGD [31].

## 5. Experiments

### 5.1. Data Collection and Preprocessing

In this paper, two publicly available datasets are used to demonstrate their generalizability, RSHA and NJUST, respectively.

RSHA [32] Dataset: The Radar Signature Dataset (RSHA) from the University of Glasgow was collected using an FMCW radar system operating at 5.8 GHz. The system features a pulse repetition period of 1 ms and a 400 MHz bandwidth, capturing 128 complex samples per scan. The dataset comprises 1754 motion captures from 72 participants, encompassing six human behaviors: walking, sitting, standing, picking up objects, drinking, and falling. Detailed information is provided in Table 1. Figure 4a–f display spectrograms of the six actions following the final preprocessing steps.NJUST [33] Dataset: This dataset was collected by the School of Electrical and Optical Engineering at Nanjing University of Science and Technology using a portable FMCW radar with a 320 MHz bandwidth, 3.3 ms frequency ramp repetition period, and +8 dBm average transmit power. A pair of 2 × 2 patch antenna arrays transmitted and received C-band signals at a height of 0.8 to 1 m. It includes six human behaviors: fall, jog, jump, squat, step, and walk. The dataset is generally balanced, with fall being slightly underrepresented. Table 2 provides details, and Figure 4g–l show spectrograms of the actions post-preprocessing.

All spectrograms, after undergoing data augmentation, are resized to 224 × 224 pixels and normalized before being fed into the network. Data augmentation includes transformations such as rotation, scaling, and noise addition to enhance the robustness of the model. For validation and testing, we randomly select 50 spectrograms per motion category, ensuring a balanced representation across different actions. The remaining images are used for training. In this semi-supervised learning model, 10% of the training samples are randomly selected and labeled, providing a small yet informative set for supervised learning. The rest of the samples remain unlabeled, enabling the model to leverage semi-supervised techniques such as pseudo-labeling and consistency regularization to learn from the abundant unlabeled data, thereby improving overall classification accuracy and generalization capabilities. This approach not only reduces the dependency on extensive labeled datasets but also enhances the model’s performance in real-world scenarios with limited labeled data.

The semi-supervised model was implemented on a server equipped with an Intel Xeon Platinum 8352V CPU (Intel, Santa Clara, CA, USA) and an NVIDIA GeForce RTX 4090 GPU (NVIDIA, Santa Clara, CA, USA), running on Ubuntu 20.04. The deployment utilized Python 3.8 and PyTorch 1.11.0 open-source software. The training hyperparameters are detailed in Table 3.

### 5.2. Ablation Experiments

To evaluate the effectiveness of the proposed improvement strategy, including the role of the shared parameter approach, we conducted ablation experiments on five models. Under the same epoch conditions, the first model employs only the pseudo-labeling (PL) method, the second model uses only contrastive learning (CL), the third model combines consistency regularization (CR) with pseudo-labeling, and the fourth model integrates contrastive learning, pseudo-labeling, and consistency regularization. The final model utilizes our proposed method.

As shown in Figure 5, the accuracy of the pseudo-labeling method alone eventually reaches 87.21%. The initial fluctuations might be due to incorrect pseudo-label generation, but the accuracy stabilizes over time with minor variations. In contrast, the accuracy for the contrastive learning method alone, although not surpassing that of pseudo-labeling, remains relatively stable and peaks at 86.23%. When combining consistency regularization with pseudo-labeling, the accuracy trend shows a gradual increase, but the efficiency is too slow. Even when other algorithms have stabilized, this method fails to achieve the best results, reaching only 77.43% in the same number of iterations. The fourth method, which combines consistency regularization, contrastive learning, and pseudo-labeling, shows some improvement in accuracy over the third method but still only achieves 85.9%, falling short of expectations. Our proposed method, which incorporates the shared parameter strategy, quickly generates correct pseudo-labels and accurately recognizes human actions. The results were remarkable. This approach achieves a final accuracy of 91.48%.

### 5.3. The Impact of the Number of Labels

To investigate the impact of different proportions of labeled radar feature maps on the accuracy of human action recognition models, we meticulously controlled the labeling ratios in our experiments. Specifically, we labeled 5%, 10%, 20%, 30%, 40%, and 50% of radar feature maps in two datasets (RSHA and NJUST) and trained the models using these labeled data. This design enabled us to systematically analyze the effect of the amount of labeled data on model performance. Figure 6 illustrates the trend of model accuracy on different datasets as the proportion of labeled data increases.

It is evident from the figure that when the proportion of labeled data increases from 5% to 10%, there is a significant improvement in model accuracy, typically in the range of 2–4%. This phenomenon indicates that a very small amount of labeled data limits the model’s ability to learn features, resulting in poor performance in human action recognition. However, when the proportion of labeled data reaches 10%, the model can better capture important features in the data, leading to a substantial performance improvement.

As the proportion of labeled data further increases to 20% and 30%, model accuracy continues to improve, but the rate of improvement starts to slow down. This suggests that within this range, the model has already learned the primary features of the data well, and additional labeled data, while still beneficial, contributes less significantly to performance enhancement. Particularly at the 30% labeling ratio, the accuracy trend becomes more stable, indicating that the model has reached a performance bottleneck.

When the proportion of labeled data reaches 50%, the model accuracy on both datasets approaches approximately 94%. This result demonstrates that once the labeled data reaches a certain scale, the model can fully utilize these data to perform more precise feature learning and classification, thereby significantly improving the accuracy of human action recognition.

### 5.4. Performance Comparison

We assessed the efficacy of our proposed approach using two publicly accessible datasets and benchmarked it against advanced networks and state-of-the-art methods. Table 4 presents the outcomes of our comparative analysis.

DenseNet employs dense connectivity, where each layer is directly connected to the outputs of all the layers before it. We trained 10% of the labeled radar feature dataset. After that, the remaining data were utilized for testing.SeResNet uses residual connections and squeeze-and-excitation (SE) blocks, which enhance the feature representation of the model by adaptively relabeling the channels. The model was trained and tested in the same way as DenseNet.Pseudo-labeling [27] is a more common type of semi-supervised learning, where we first pretrained the VGG19 using the ImageNet dataset, followed by fine-tuning using the 10% labeled radar feature dataset. Afterwards, pseudo-labels were generated based on the fine-tuned VGG19 model, and then the 10% labeled radar feature dataset was used for training.MocoV3 [34] is a contrastive semi-supervised learning method that has an important place in contrastive learning. In this algorithm, the radar feature map dataset was utilized first for training, and after that the dataset was tested.FixMatch [14] is a classic semi-supervised model in recent years. We used the dataset with 10% band labels for training to generate pseudo-labels. Finally, it was tested. Since FixMatch has a hard time converging in the same number of iterations, we increased the number of training iterations until it finishes converging.JDS-TL [18] is a transfer learning algorithm that combines an unsupervised adversarial domain transfer module with a supervised semantic transfer module. It focuses on training HAR models using sparsely labeled datasets.AA-CAE [16] is a semi-supervised method that initially utilizes unsupervised pretraining to initialize the network, followed by supervised fine-tuning.

When randomly selecting 10% labeled data from the training dataset, all methods were repeated five times to ensure consistency in accuracy. Table 4 presents results for two different sensing tasks, where we evaluate the performance of MF-Match using the rate of correct predictions as a metric. Under specific dataset conditions, due to the scarcity of labeled data, training networks alone yielded lower accuracy rates, especially DenseNet, which achieved only around 50% accuracy using dense connectivity for learning. In contrast, other methods benefited significantly from a large amount of unlabeled data, achieving accuracy rates above 85% for the RSHA dataset and around 80% for the NJUST dataset. Among them, AA-CAE, pseudo-labels, and FixMatch demonstrated strong learning capabilities, achieving accuracy rates close to 90%. However, AA-CAE and pseudo-labels required pretraining followed by fine-tuning, while FixMatch exhibited slower learning, requiring iterative improvements to achieve a high-quality model for prediction. In contrast, our proposed method does not necessitate pretraining or class balancing, delivering effective results in fewer iterations, making it advantageous for practical applications.

### 5.5. Classification Results

Figure 7 shows the confusion matrix of the dataset in the proposed MF-Match algorithm. This matrix is used to validate the proposed SSL method with limited labeled data. To prevent data leakage, we tested the radar feature maps of individuals numbered P58 to P72, who were not involved in the training process. The diagonal of the confusion matrix represents the individual classification accuracy for each action.

As shown in Figure 7, the model performs exceptionally well in recognizing the actions Fall Down and Walk. The precision, recall, specificity, and F1-score are all nearly 1.0, indicating that the model can accurately identify almost all instances of these two actions. For the Sit action, both the precision and specificity are 1.0, while the recall is 0.94 and the F1-score is 0.97. This demonstrates that the model can very accurately recognize the Sit action and effectively avoid misclassifying other actions as Sit. For the Stand action, the precision and specificity are both 1.0, with a recall of 0.86 and an F1-score of 0.92. Although the recall is slightly lower, the model’s overall performance in recognizing the Stand action remains high. Lastly, for the Drink action, the precision is 0.88 and the F1-score is 0.84, which are lower compared to other actions. Despite a lower recall of 0.8, the specificity is relatively high at 0.96, indicating that the model can correctly identify non-Drink actions in most cases. We attribute this performance to the fact that Pick Up and Drink are both in situ nonperiodic motions and exhibit certain similarities in the feature maps. Finally, the accuracy of MF-Match on the NSHA dataset was calculated to be 0.91.

### 5.6. Complexity Analysis

In this section, we examine the efficiency of the proposed framework. Both MF-Match and FixMatch utilize consistency regularization methods and are end-to-end semi-supervised learning frameworks. We compared the floating-point operations (FLOPs), total training time, and testing time of our framework with the widely used semi-supervised model FixMatch, as shown in Table 5. Compared to FixMatch, MF-Match’s FLOPs increased by 39%. We attribute this primarily to the requirement of a separate module for contrastive learning when handling pseudo-labels, which enhances the quality of pseudo-labels and consequently improves model accuracy. This improvement in pseudo-label accuracy also aids in the latter stages of training and convergence of the model. Although the total training time appears significantly different, the difference in training time for one epoch is not significant. The main reason for the substantial difference in total time is that FixMatch is misled by incorrect pseudo-labels. However, our model, which incorporates a contrastive learning module and parameter-sharing strategies, accurately guides the annotation of pseudo-labels, accelerating model convergence. Consequently, the number of iterations required is considerably fewer, leading to a shorter overall training time.

### 5.7. Application of VIT

In recent years, the Vision Transformer (ViT) has been rapidly developed with the use of Transformers in vision-based modeling. The architecture of the ViT is based on Transformers, but instead of accepting only sequential inputs as normal Transformers do, the input image is segmented into small, nonoverlapping patches and they are projected into the patch embedding. After that, a one-dimensional learnable positional encoding is added to the patch embedding to preserve the spatial information, and finally the joint embedding is fed to the encoder.

Therefore, during our experiments, we also tried to use a ViT. Figure 8 shows the comparison between our model and the results after using the ViT. In this paper, the classical residual network is used.

Figure 8 shows that the Vision Transformer (ViT) underperforms compared to the residual network, achieving an accuracy of 89.8%, which is nearly 2% lower than our proposed model. This result contrasts with the findings in [35], prompting an analysis of potential causes. Firstly, the position embedding in ViTs may not be suitable for radar images. Unlike conventional RGB images, radar images have distinct spatial structures and lack clear semantic features, making it difficult for ViTs to identify significant local features across different categories. Radar image features rely more on local spatial relationships. Secondly, in contrast to larger datasets like CIFAR-100 or CIFAR-10, the RSHA dataset is relatively small. This limited data restricts the ViT from acquiring sufficient prior knowledge about the images, resulting in inferior performance compared to the residual network.

## 6. Conclusions

This study introduces MF-Match, a novel architecture for semi-supervised HAR using radar spectrograms. The proposed approach significantly reduces the requirement for extensive radar signal labeling by leveraging small amounts of labeled data in conjunction with a large volume of unlabeled data. A semi-supervised framework is adopted to extract discriminative features from unlabeled data, thereby minimizing the reliance on labeled samples. To address the challenge of distinguishing between different behavioral signals, a comparative learning-based pseudo-label generation method is proposed. This method enhances inter-class feature distinctions through multiple augmentation strategies, thereby improving recognition accuracy. The effectiveness of MF-Match is demonstrated on two publicly available radar feature map datasets with 10% labeled samples. The proposed approach achieves recognition accuracies of 91.48% and 86.69%, respectively, surpassing existing methods. Ablation experiments confirm the efficacy of the proposed approach, demonstrating superior performance over seven state-of-the-art methods in radar-based HAR scenarios with limited labeled data. Ongoing optimizations are being conducted to further enhance the classification performance of the proposed system.

## Figures and Tables

**Figure 1 sensors-24-04940-f001:**
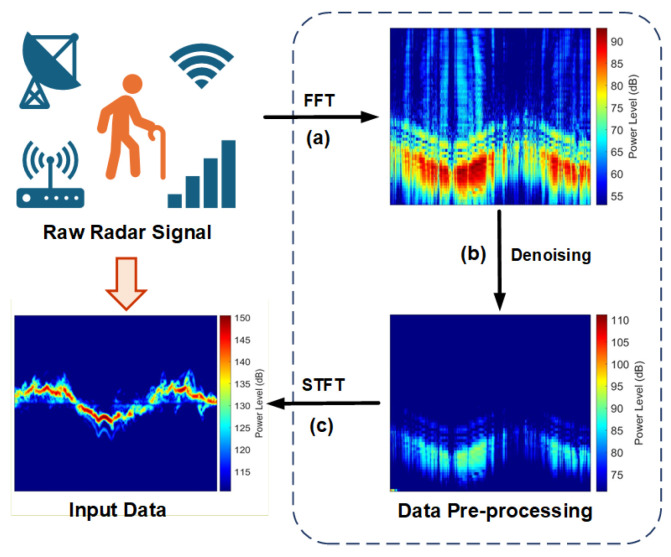
Processing sequence of radar returns: (**a**) fast Fourier transform (FFT). (**b**) Elimination of low-frequency noise. (**c**) Short-time Fourier transform (STFT) and normalization.

**Figure 2 sensors-24-04940-f002:**
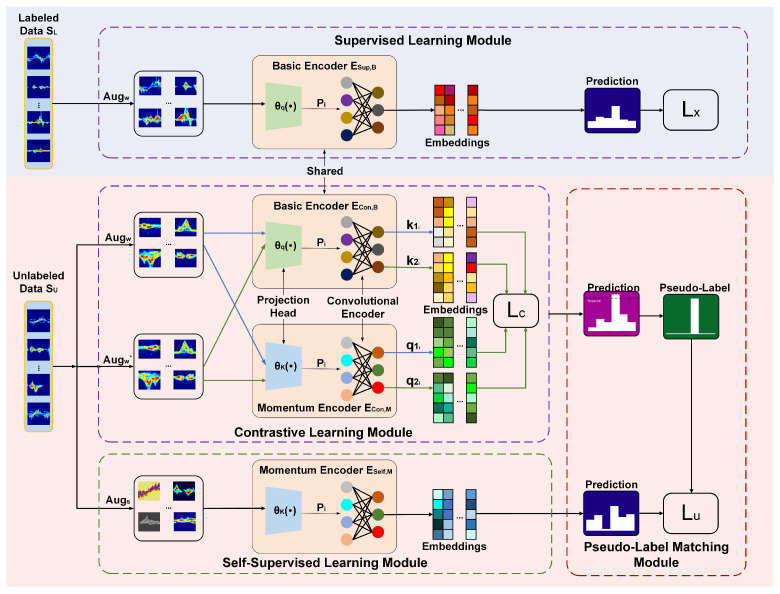
MF-Match structure diagram.

**Figure 3 sensors-24-04940-f003:**
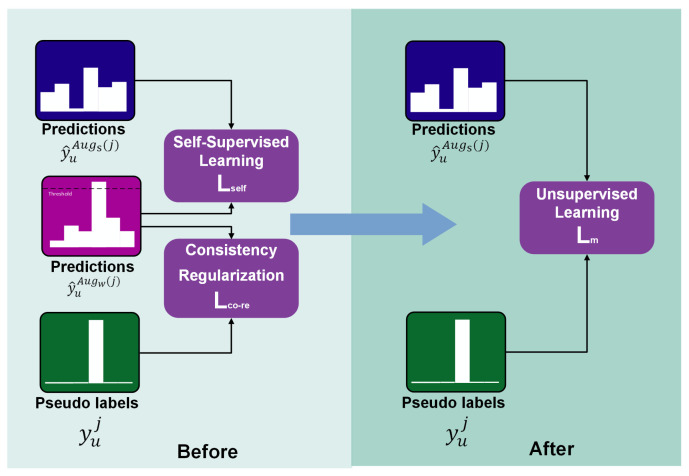
Loss function integration.

**Figure 4 sensors-24-04940-f004:**
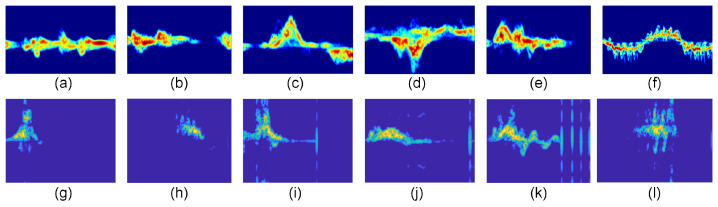
Micro-Doppler characteristics of typical samples of the dataset. (**a**–**f**) RSHA dataset. (**a**) Drink water. (**b**) Fall down. (**c**) Pick up something. (**d**) Sit down. (**e**) Stand up. (**f**) Walk. (**g**–**l**) NJUST dataset. (**g**) Fall. (**h**) Jog. (**i**) Jump. (**j**) Squat. (**l**) Walk.

**Figure 5 sensors-24-04940-f005:**
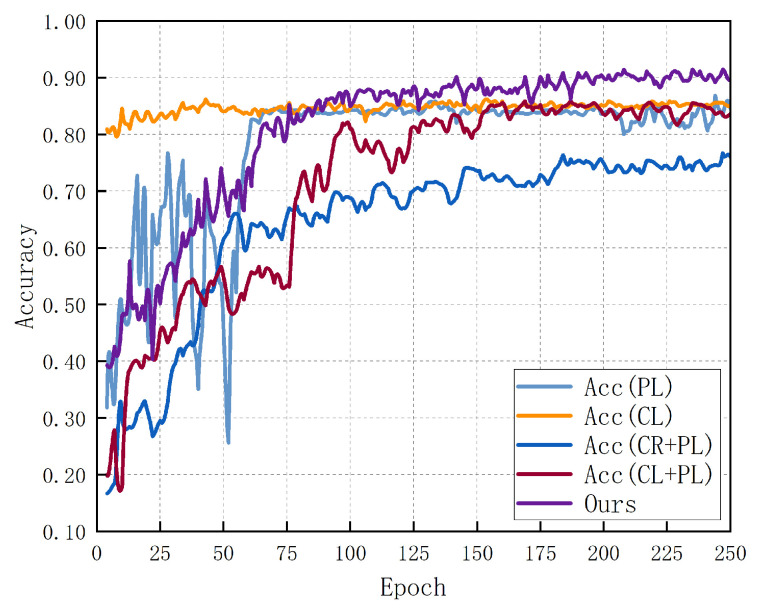
Results of ablation experiments.

**Figure 6 sensors-24-04940-f006:**
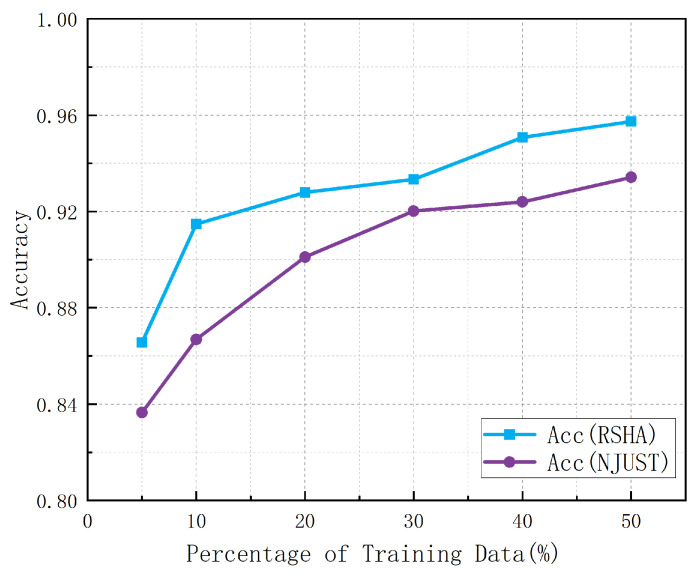
Comparison of network structures.

**Figure 7 sensors-24-04940-f007:**
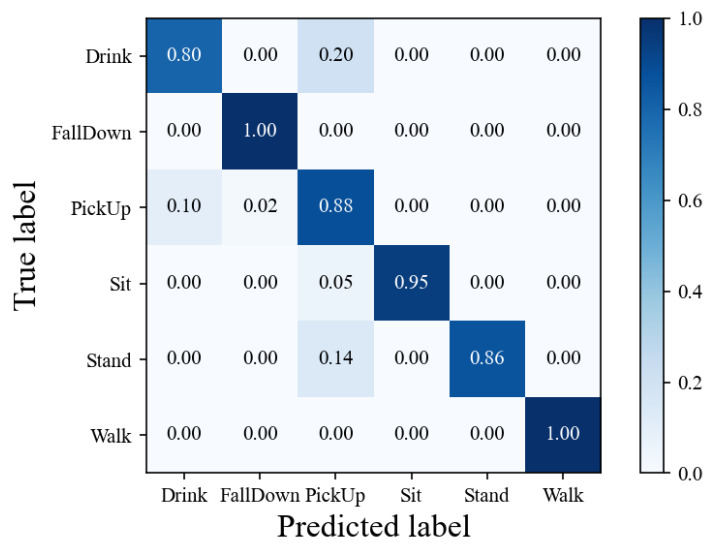
Confusion matrix.

**Figure 8 sensors-24-04940-f008:**
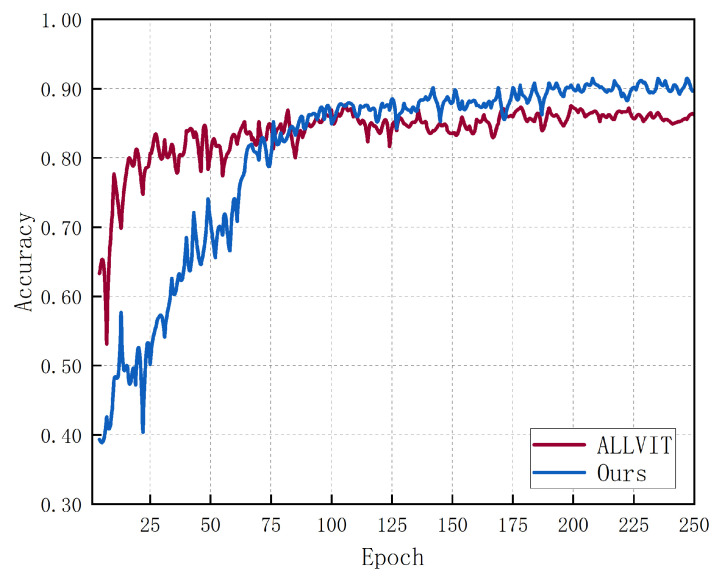
Comparison of network structures.

**Table 1 sensors-24-04940-t001:** Summary of RSHA datasets.

No.	Description of Activities	Sample Size
A1	Drink water	310
A2	Fall down	198
A3	Pick up something	311
A4	Sit down	312
A5	Stand up	311
A6	Walk back and forth	312

**Table 2 sensors-24-04940-t002:** Summary of NJUST datasets.

No.	Description of Activities	Sample Size
A1	fall	209
A2	jog	239
A3	jump	240
A4	squat	238
A5	step	239
A6	walk	240

**Table 3 sensors-24-04940-t003:** Training hyperparameters.

Parameters	Value	
Input size	224 × 224	
Label batch size	6	
Unlabel batch size	24	
Learning rate	Classifier:	4.00 × 10^−4^
	Decay rate:	5.00 × 10^−4^
Weight decay	5 × 10^−4^	
Epoch	250	
Threshold	0.9	
Optimizer	Gradient descent (SGD)	

**Table 4 sensors-24-04940-t004:** Comparison of accuracy performance with 10% labeled data.

Method	Labeled Data (10%)	Unlabeled Data	RSHA ACC	NJUST ACC
DenseNet	Yes	No	53.80%	45.10%
SeResNet	Yes	No	77.00%	74.13%
Pseudo-labels(VGG19)	Yes	Yes	89.51%	85.93%
MocoV3	Yes	Yes	86.30%	77.08%
FixMatch	Yes	Yes	87.83%	84.72%
JDS-TL	Yes	Yes	87.60%	85.89%
AA-CAE	Yes	Yes	89.47%	84.72%
Ours	Yes	Yes	91.48%	86.69%

**Table 5 sensors-24-04940-t005:** FLOPs, total training time, and testing time of FixMatch and MF-Match.

Method	FLOPs (G)	Total Training Time (s)	Testing Time (s)
FixMatch	10.57	45,000	2
MF-Match	14.7	32,750	2

## Data Availability

This article mainly uses two publicly available datasets, the first being the RSHA dataset produced by the University of Glasgow, located at https://researchdata.gla.ac.uk/848/ (accessed at 1 January 2024). The second dataset is the NJUST dataset produced by the School of Electronic and Optical Engineering, Nanjing University of Science and Technology, located at https://ieee-dataport.org/documents/human-activity-data-58ghz-fmcw-radar (accessed at 24 May 2024).

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
