# Peer review of "MF-Match: A Semi-Supervised Model for Human Action Recognition"

_sensors, 2024, doi:10.3390/s24154940_

Round 1

Reviewer 1 Report

Comments and Suggestions for Authors

This paper presents a semi-supervised learning framework for human action recognition using radar signals. The proposed framework is novel and the topic is interesting. However, the following issues need to be addressed before publication:

1.       In the proposed framework, the model parameters are learned from scarce labeled data and shared with the learning of abundant unlabeled data. The potential risk of overfitting needs to be analyzed and discussed both theoretically and empirically. The ablation study is oversimplified.

2.       The proposed framework includes several modules, but the ablation study only focuses on the parameter-sharing part.

3.       The performance comparison is not satisfactory, as only scenarios with 10% of labeled data are considered. Please present results for more scenarios.

4.       The correct use of terminology should be checked. For example, there are several “versions” of “contrastive learning” mentioned in this manuscript.

5.       Section 5.2 is redundant.

Comments on the Quality of English Language

The overall writing is satisfactory, but make sure to use the right terminologies.

Reviewer 2 Report

Comments and Suggestions for Authors

The authors propose a semi-supervised learning algorithm, MF-Match, which computes the pseudo-labels of larger-scale unsupervised radar data, assisting in improving the accuracy of HAR algorithms. While the approach has certain advantages from a methodological perspective, there are several significant issues that need to be addressed and clarified. Below are the specific comments:

Major Concerns:

1. Methodology:

The authors mention two weight parameters in the loss function section. How exactly are these parameters set?

2. Experiments:

   The authors used publicly available datasets for experiments, which is commendable. However, there are some issues in the experimental setup that need to be discussed:

     1. The authors describe using a 95% window overlap in their experiments. This is a bold choice. Please explain the rationale behind this setting, as such a large overlap could significantly degrade the time resolution of the STFT.

     2. The authors state that 10% of the data was randomly selected for training. Given the significant differences in actions performed by different subjects, this random selection method could lead to data leakage, where some subjects in the test set have been seen in the training set. This would result in inflated performance metrics. The best practice would be to use leave-one-subject-out cross-validation.

     3. Why was a 256-point FFT chosen for the FFT processing, and not another point size?

     4. The authors used an IIR filter to remove low-frequency noise. Why specifically low-frequency noise? How was the filtering range determined? How did the authors consider these issues?

3. Results:

  1. The experimental results were not sufficiently discussed. The authors omitted crucial parts of the confusion matrix. Please provide specific results and discuss them.

  2. Comparing only the accuracy is not enough. Please also provide comparisons of time complexity and model complexity and explain them.

  3. The content of Table 4 is insufficient. Please add comparisons with other related work, especially recent studies.

Minor Concerns:

1. There are numerous grammatical errors throughout the paper that need special attention and correction.

2. The figures and tables in the paper need improvement. For example, the text in Figure 3 is too small, and there is too much white space; Figure 2 is too small; the text in Figure 1 is too large; Figures 5 and 6 are not aesthetically pleasing.

Comments on the Quality of English Language

Extensive editing of English language required

Round 2

Reviewer 1 Report

Comments and Suggestions for Authors

My concerns have been addressed. I have no additional questions/confusions on this paper. I am positive to its publication.

Reviewer 2 Report

Comments and Suggestions for Authors

The author has addressed all my concerns

Comments on the Quality of English Language

Extensive editing of English language required